# Anomalous excitonic phase diagram in band-gap-tuned Ta₂Ni(Se,S)₅

Cheng Chen[1,2,13], Weichen Tang [3,4,13], Xiang Chen [3,4], Zhibo Kang[2], Shuhan Ding[5], Kirsty Scott[6], Siqi Wang[2], Zhenglu Li [3,4,7], Jacob P. C. Ruff[8], Makoto Hashimoto [9], Dong-Hui Lu [9], Chris Jozwiak [10], Aaron Bostwick [10], Eli Rotenberg[10], Eduardo H. da Silva Neto [6], Robert J. Birgeneau [3,4,11], Yulin Chen [1], Steven G. Louie [3,4] ✉, Yao Wang [5,12] ✉ & Yu He [2] ✉

During a band-gap-tuned semimetal-to-semiconductor transition, Coulomb attraction between electrons and holes can cause spontaneously formed excitons near the zero-band-gap point, or the Lifshitz transition point. This has become an important route to realize bulk excitonic insulators – an insulating ground state distinct from single-particle band insulators. How this route manifests from weak to strong coupling is not clear. In this work, using angle-resolved photoemission spectroscopy (ARPES) and high-resolution synchro-tron x-ray diffraction (XRD), we investigate the broken symmetry state across the semimetal-to-semiconductor transition in a leading bulk excitonic insu-lator candidate system Ta₂Ni(Se,S)₅. A broken symmetry phase is found to be continuously suppressed from the semimetal side to the semiconductor side, contradicting the anticipated maximal excitonic instability around the Lifshitz transition. Bolstered by first-principles and model calculations, we find strong interband electron-phonon coupling to play a crucial role in the enhanced symmetry breaking on the semimetal side of the phase diagram. Our results not only provide insight into the longstanding debate of the nature of inter-twined orders in Ta₂NiSe₅, but also establish a basis for exploring band-gap-tuned structural and electronic instabilities in strongly coupled systems.

When the indirect band gap of a material is continuously reduced from positive to negative, band theory predicts a semiconductor-to-semimetal Lifshitz transition. However, such a transition may never be reached in the presence of electron-hole Coulomb attraction. It is postulated that poor screening in low carrier density semimetals and strong electron-hole binding in narrow gap semiconductors can both

result in the spontaneous formation of excitons[1,2]. The condensation of excitons results in an insulating ground state, the so-called excitonic insulator (EI), which is separated from the normal state by a dome-shaped phase boundary peaked around the Lifshitz point (see Fig. 1a)[3]. This simple setup became a primary platform to search for condensed excitons and other exotic correlated phases, as well as potential solid-

[1]Department of Physics, University of Oxford, Oxford OX1 3PU, United Kingdom. [2]Department of Applied Physics, Yale University, New Haven, CT 06511, USA. [3]Physics Department, University of California, Berkeley, CA 94720, USA. [4]Materials Sciences Division, Lawrence Berkeley National Lab, Berkeley, CA 94720, USA. [5]Department of Physics and Astronomy, Clemson University, Clemson, SC 29631, USA. [6]Department of Physics, Yale University, New Haven, CT 06511, USA. [7]Mork Family Department of Chemical Engineering and Materials Science, University of Southern California, Los Angeles, CA 90089, USA. [8]Cornell High Energy Synchrotron Source, Cornell University, Ithaca, NY 14853, USA. [9]Stanford Synchrotron Radiation Lightsource, SLAC National Accelerator Laboratory, Menlo Park, CA 94025, USA. [10]Advanced Light Source, Lawrence Berkeley National Laboratory, Berkeley, CA 94720, USA. [11]Department of Materials Science and Engineering, University of California, Berkeley, CA 94720, USA. [12]Department of Chemistry, Emory University, Atlanta, GA 30322, USA. [13]These authors contributed equally: Cheng Chen, Weichen Tang. ✉e-mail: sglouie@berkeley.edu; yao.wang@emory.edu; yu.he@yale.edu

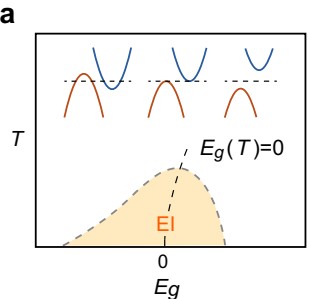

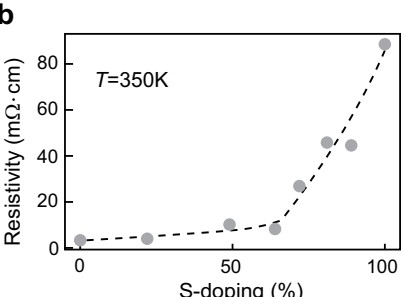

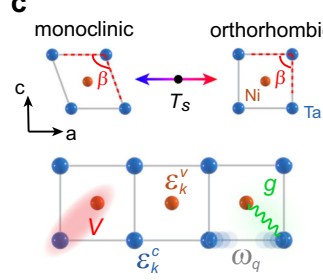

**Fig. 1 | Band-gap-tuned semimetal-to-semiconductor transition in narrow band gap systems. a** Archetypal phase diagrams in indirect band gap controlled excitonic insulator (EI) across the semimetal-to-semiconductor transition. **b** Normal state electrical resistivity evolution with S-doping in $Ta_2Ni(Se,S)_5$. **c** Structure motif of quasi-1D $Ta_2Ni(Se,S)_5$ system below and above transition temperature $T_s$ (upper panel). The lattice distortion in the monoclinic phase is exaggerated. Symbolic sketch of a minimal lattice model of the system used to describe the low-energy interactions in the system (lower panel). $\epsilon_k^c$ and $\epsilon_k^v$ are kinetic energies of conduction and valence electrons. $\omega_q$ is the chain-shearing $B_{2g}$ phonon frequency. $g$ is the interband electron-phonon coupling vertex. $V$ is the electron-hole Coulomb interaction potential.

state BCS-BEC crossover phenomena[4–7]. For example, adding electronic anisotropy can further narrow the EI dome due to reduced Coulomb attraction[8,9], and sufficiently heavy holes may cause Wigner crystallization[10]. Moreover, band-gap-tuned BCS-BEC crossover has been recently suggested in 2D moiré systems[11]. However, interband electron-phonon interaction can also lead to a gapped charge-density-wave state that occurs concurrently with the exciton condensation[4]. It remains unclear how the excitonic instability and the density-wave instability interact with each other, and whether it is possible at all to distinguish them. Since no super-transport (charge or heat) is associated with the excitonic insulator phase, and its analogy to BCS superconductivity is only formal[10], an effective experimental method to distinguish the Coulomb and lattice channels is to identify a tuning parameter that gives differentiating predictions of the phase diagram.

Band-gap-controlled materials that undergo semimetal-to-semiconductor transitions provide an ideal platform to investigate this issue. However, the experimental realization of such a transition in bulk systems has been challenging. Divalent metals under hydrostatic pressure and uniaxial strain were first proposed as promising candidates[12], but experimental evidence was indirect and scarce without clear ways to separate Coulomb and electron-phonon interaction effects. Recent experiments suggest a potential exciton condensation in $1T$-TiSe$_2$[13], where the band gap tunability is limited. Quasi-one-dimensional (quasi-1D) ternary chalcogenide $Ta_2NiSe_5$ has recently emerged as another leading EI candidate, exhibiting an orthorhombic-to-monoclinic structural transition at $T_s = 329$ K, concomitant with a putative exciton condensation[14,15]. The system consists of parallel quasi-1D Ta and Ni chains (see Fig. 1c), where long-lived electron-hole pairs are supposed to dwell on Ta $5d_{x^2-y^2}$ and Ni $3d_{xy}$ orbitals. On the one hand, the sizable ground-state single-particle gap[16–19], minute structural distortion[14,20], and a "dome-like" temperature-pressure phase diagram[10,15] are all ostensibly consistent with the predictions in a band-gap-tuned EI. On the other hand, strong electron-phonon coupling effects are also found to coexist with correlation effects[20–32]. With the presence of both interactions, the band structure alone no longer sufficiently dictates the excitonic insulator phase region, and it is unclear how the low-temperature broken symmetry state would evolve across the Lifshitz transition point. In $Ta_2Ni(Se,S)_5$, this question can be addressed since the value of the direct band gap can be effectively tuned by isovalent sulfur substitution of selenium[15,33]. However, the phase diagram of $Ta_2Ni(Se,S)_5$ – especially where the broken symmetry phase ends and where the normal state crosses from a gapless to a gapped electronic structure with S-doping – has seen vastly contradicting results[14,15,27], mainly due to the lack of direct probes of the single-particle band gap and the order parameter of the broken symmetry.

In this work, using angle-resolved photoemission spectroscopy (ARPES) and high-resolution synchrotron x-ray diffraction (XRD), we directly probe the electronic and lattice structure of $Ta_2Ni(Se,S)_5$ system and map out its phase diagram across the semimetal-to-semiconductor. A broken symmetry phase is found to be continuously suppressed from the semimetal side to the semiconductor side, contradicting the anticipated maximal excitonic instability around the Lifshitz transition. Bolstered by first-principles and model calculations, we reveal the crucial role of strong interband electron-phonon coupling behind the phase diagram.

## Results

The general impact of S-substitution on the low-energy electronic structure can be revealed through the normal-state (defined here as the high-temperature symmetric structure phase) resistivity measured along the chain direction, which slowly increases until an abrupt upturn around 70% doping (see Fig. 1b). Such an upturn indicates a potential Lifshitz transition on the normal-state band structures. Hence, S-doped $Ta_2NiSe_5$ constitutes an ideal platform to study band-gap-tuned semimetal-to-semiconductor transition in an electron-phonon coupled correlated system. We first track the evolution of the broken symmetry phase boundary in $Ta_2Ni(Se,S)_5$ with direct measurements of the structural order parameter. In bulk excitonic insulators, exciton condensation is always tied to a simultaneous lattice distortion[10]. Employing high-resolution single-crystal synchrotron XRD, Fig. 2a characterizes the temperature- and S-substitution-dependent evolution of the lattice order parameter $\beta$ (also see Supplementary Note 1), which is defined as the angle between the lattice $a$ and $c$ axes shown in Fig. 1c. We find the structural phase transition remains second-order throughout the entire S-doping range. The ground-state order parameter $\beta(T=0)$ monotonically decreases with increasing S-doping (Fig. 2b), and the structural transition temperature only approaches zero near full S-substitution ($Ta_2NiS_5$) (bottom panel in Fig. 2b). This is notably different from the electronic phase diagram inferred from resistivity (see Fig. 2b and Supplementary Note 1), which instead suggests an abrupt collapse of the broken symmetry phase at an intermediate S-doping level[15]. Therefore, it is imperative to obtain a direct electronic structure view to determine the normal state semimetal-to-semiconductor transition point $p_L$ along the S-doping axis.

Previous studies have reported conflicting values of $p_L$ from resistivity, Raman measurements[15,27]. Here, we conduct high-resolution ARPES measurement, which is a direct probe of electronic structure, to reveal the evolution of band overlap/gap of $Ta_2Ni(Se,S)_5$ across the semimetal-semiconductor phase transition. The high statistics of the measurements enable electronic state restoration up to $4.5\,k_BT$ above

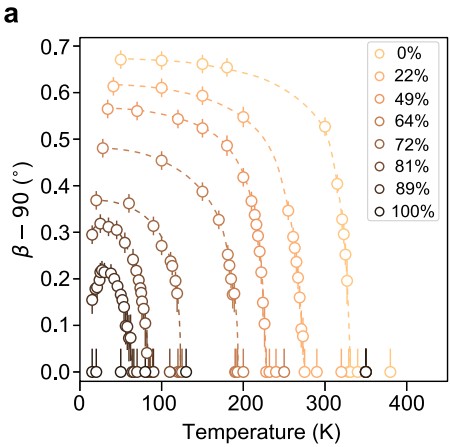

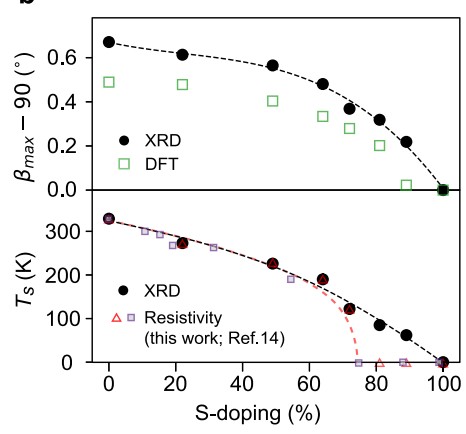

**Fig. 2 | Structural order parameter of Ta₂Ni(Se,S)₅. a** Change of the inter- and intra-chain Ta-Ta bond angle $\beta - 90°$ as a function of temperature and S-doping. Error bars are determined based on the fitting error of the Bragg peak positions.

**b** Top: experimental and DFT calculated $\beta - 90°$ at zero temperature. Bottom: structural transition temperature $T_s$ as a function of S-doping level, determined from XRD and resistivity measurement.

$E_F$ (see Methods), and the orbital selectivity through polarized X-ray beam, allows the direct comparison with density functional theory (DFT) calculations, both of which help us precisely locate $p_L$. Figure 3 shows the photoemission spectra in both the normal and the broken symmetry state (also see Supplementary Note 1). As the S-doping level increases, the Ta $5d_{x^2-y^2}$ orbital component (blue) gradually withdraws from the valence band top, which becomes dominated by the Ni $3d_{xy}$ orbital (red). Pseudogap states, the strong depletion of spectra intensity around Fermi level[19,20], are also observed, which account for the high-temperature insulating behavior in these semimetallic samples (see ref. 20 for detailed description). Beyond $p_L$ ( ~ 70% S-doping), the normal-state conduction and valence bands are completely separated in energy (see Fig. 3a) and, simultaneously in the broken-symmetry-state, the Ta orbital component largely disappears from the valence band top (see Fig. 3b). Fitting the normal-state band dispersions (see Supplementary Note 2 for details) quantifies the band gap (or band overlap in the case of negative values), which changes sign across $p_L$ (Fig. 3c). In addition, revealed through the "M-shape" dispersion of the valence band top in the broken symmetry state, the band back-bending momentum $k_F$ also approaches Γ near $p_L$ (Fig. 3d). Both results are consistent with a concurrent normal state semimetal-to-semiconductor transition. Our DFT results further confirm such a Lifshitz transition near $p_L$ via the band-gap-sign flip and the $k_F$ reduction at the single-particle level, without involving ad hoc electron-hole Coulomb attraction (see Fig. 3c, d and Supplementary Note 3).

The ARPES measurement of the semimetal-to-semiconductor transition doping is consistent with that deduced from resistivity measurement (Figs. 1b and 2b). However, the structural phase transition temperature $T_s$ and the structural order parameter (approximated as $\beta_{max} - 90°$) are found to decrease monotonically with increasing S-doping through our XRD measurements (Fig. 2b). This poses a marked departure from the archetypal band-gap-tuned excitonic insulator phase diagram, where the excitonic instability maximizes around $p_L$ (Fig. 1a). To highlight this contrast, Fig. 4a shows the evolution of both the measured structural transition temperature $T_s$ and the experimentally determined Ta atom displacement $|X|$ in Ta₂Ni(Se,S)₅. Both $|X|$ and $T_s$ appear insensitive to the system crossing $p_L$, which leads to considerations beyond weak electron-hole Coulomb interaction. One possibility is a heavy-hole Wigner crystal phase[10]; but the electron and hole effective masses are both near the free electron mass in Ta₂Ni(Se,S)₅[33]. Alternatively, strong Coulomb interaction may push the phase boundary towards the semiconductor side[34], but the Hartree component will undo this effect[20] (see also Supplementary Note 4). Considering the fact that strong electron-phonon coupling

effect is revealed in both equilibrium and ultrafast pump-probe experiments in Ta₂Ni(Se,S)₅[20–28], we hereby investigate an experimentally informed minimal model with strong electron-phonon coupling, shown in Fig. 1c and Eq. (1) of Methods. This model contains both electron-hole Coulomb interactions and an interband electron-phonon coupling, which can well describe both the spectral properties of Ta₂NiSe₅ below and above $T_s$[20]. To examine the impact of strong electron-phonon coupling on the band-gap-tuned transition as a matter-of-principle question, we first set the inter-band Coulomb interaction to zero. The role of this Coulomb interaction primarily lies in a Hartree shift between two bands and can be mapped to a correction of $E_g$ (see detailed discussions in Supplementary Note 4). The average phonon displacement $|x_{ph}|$ at $T = 0$ is employed to approximate the order parameters $|X|$ measured in the experiment (Fig. 4a). With the quantized phonon modes, a zero-point fluctuation leads to the zero-coupling baseline at $|x_{ph}| = 1$.

Solved with exact diagonalization, $|x_{ph}|$ for various $E_g$ and coupling strengths are shown in Fig. 4b. In the weak-coupling case (e.g. $g = 14$ meV), the lattice order parameter displays a dome-like structure similar to the EI phase diagram derived based on Fermi-surface instability in Fig. 1a. With increasing coupling strengths, the dome gradually evolves into a monotonic crossover. This evolution exists not only for the many-body measure of fluctuations, but also for order parameters when symmetry breaking is allowed (see Supplementary Note 4). The monotonic crossover driven by strong, non-perturbative interactions reflects the irrelevance of Fermi-surface instability (top panel in Fig. 4c). The coupling strength in Ta₂NiSe₅, estimated from both ARPES spectral fitting and first-principles deformation potentials[20], is 50 ~ 60 meV, placing it in the strong-coupling regime. While more complicated models with multiple bands may be necessary to quantitatively reproduce the order parameter evolution, strong electron-phonon coupling qualitatively describes the evolution of the broken-symmetry phase across $p_L$ in experiments. Moreover, strong coupling also expands the momentum range over which the lattice becomes unstable[35], opening up the opportunity for a $\mathbf{q} = 0$ transition when $2k_F$ is small.

We demonstrate that strong electron-phonon interaction can greatly enhance the structural symmetry breaking in a semimetal, and the evolution of the broken-symmetry phase boundary can be indifferent to the semimetal-to-semiconductor transition. On the specific debate over the nature of the symmetry breaking in Ta₂Ni(Se,S)₅ systems, our results suggest a substantial deviation from the electron-hole Coulomb attraction-based EI phase (bottom panel in Fig. 4c). Instead, a strong electron-phonon-coupling induced structural

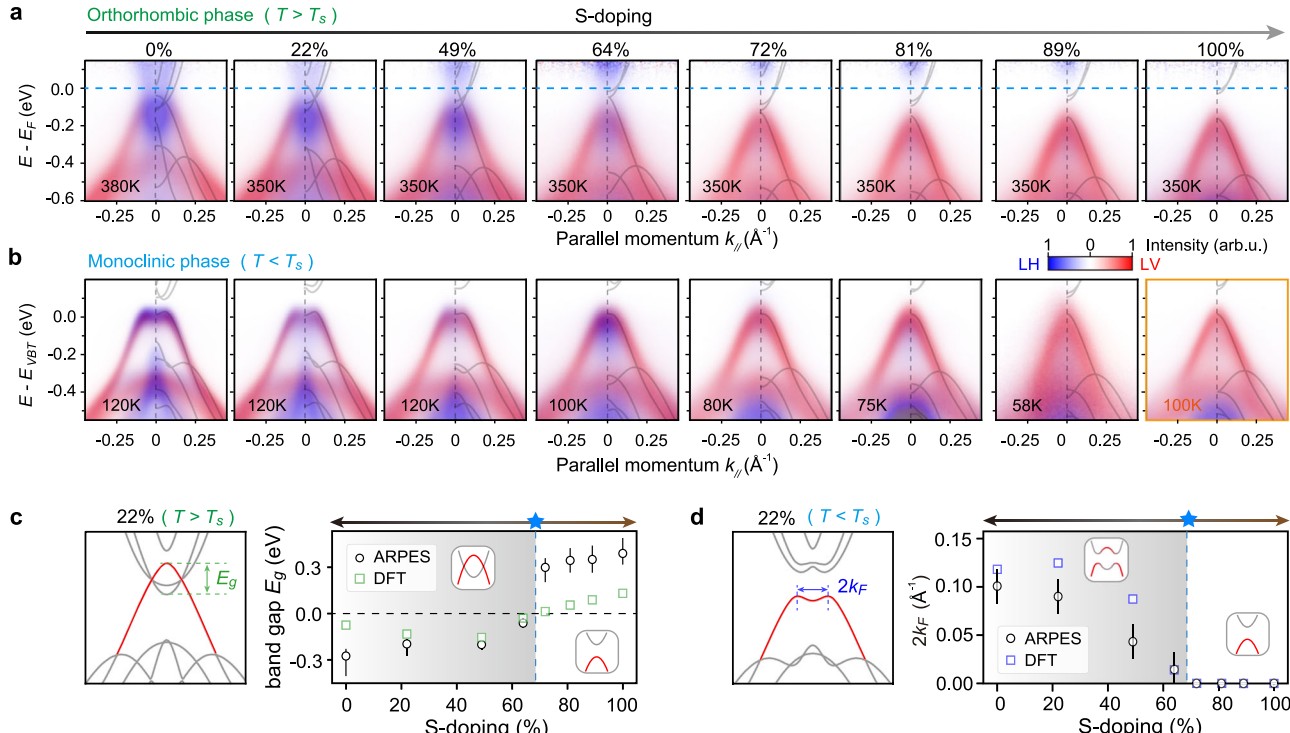

**Fig. 3 | Electronic Lifshitz transition in Ta$_2$Ni(Se,S)$_5$.** ARPES spectra along $X - \Gamma - X$ direction for samples with different S-doping levels. **a** High-temperature orthorhombic phase. **b** Low-temperature monoclinic phase (except for 100% S-doping, where no structural transition was evidenced down to 10 K). Spectra from linear horizontal (LH, blue) and linear vertical (LV, red) polarization of incident light are overlaid, highlighting Ta $5d_{x^2-y^2}$ conduction band and Ni $3d_{xy}$ valence bands respectively. VBT - valence band top. **c** DFT calculation of 22% S-doped compound

in high-temperature orthorhombic phase, and the evolution of $E_g$ extracted from DFT calculation and photoemission spectra as a function of S-doping. **d** DFT calculation of 22% S-doped compound in low-temperature monoclinic phase, and the evolution of $2k_F$ extracted from DFT calculation and photoemission spectra as a function of S-doping. Error bars are determined based on (**c**) the variation between dispersion fitting models and (**d**) the fitting error of the MDC peak position.

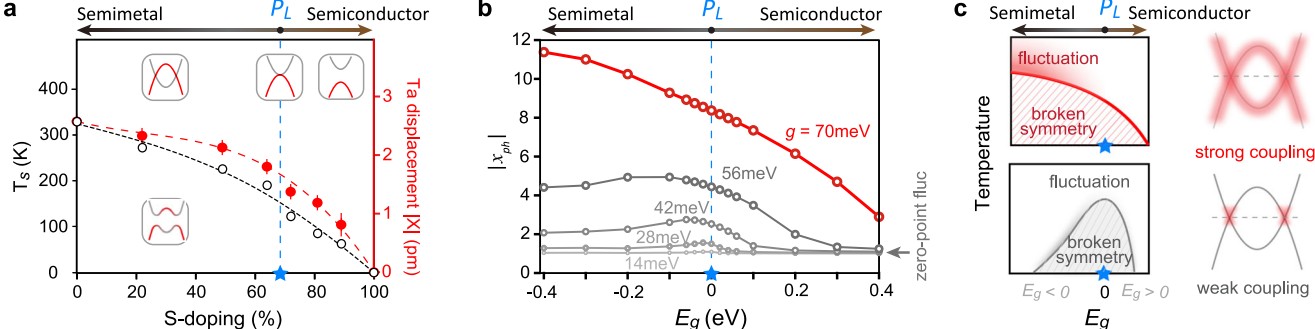

**Fig. 4 | Band gap tuned phase diagram with electron-phonon coupling.**
**a** Evolution of structural transition temperature $T_s$ and the maximum Ta atom displacement $|X|$ as a function of S-doping in Ta$_2$Ni(Se,S)$_5$. The electronic band structures within different regions are illustrated in the insets. Error bars are determined based on the fitting error of the Bragg peak positions. **b** The calculated average lattice displacement $|x_{ph}| = \sqrt{\langle \hat{x}_{ph}^2 \rangle}$ as a function of the band gap $E_g$ at various electron-phonon coupling strengths. In the quantized phonon model, the

zero-point fluctuation gives a finite $|x_{ph}| = 1$ in the decoupled limit. **c** Sketched phase diagrams of band-gap-tuned semimetal-to-semiconductor transition in the strong (top) and weak (bottom) coupling limits. Solid lines mark the phase boundaries, above which the shades show the order fluctuations[20]. Red shade on the bands illustrates the electronic states that participate in the phase transition. The horizontal dashed lines denote $E_F$.

transition can explain the observed symmetry breaking. This conclusion is consistent with recent experimental evidence of the crucial role of phonons in the equilibrium and nonequilibrium properties of Ta$_2$NiSe$_5$[22,23]. Meanwhile, it has been suggested recently that additional Coulomb and acoustic-phonon effects can also cooperatively contribute to the transition[27]. On the high-temperature fluctuations, previous investigations have identified a normal-state spectral "pseudogap", characterized by an anomalous spectral weight accumulation near the conduction band bottom. This phenomenon is

interpreted as the signature of preformed excitons due to strong electronic correlation or electron coupling to lattice fluctuations[20,36]. We find that such a spectral anomaly ubiquitously exists in the normal state on the semimetal side of Ta$_2$Ni(Se,S)$_5$, but rapidly disappears on the semiconductor side (Fig. 3a). Intriguingly, the boundary of the pseudogap, or preformed excitons, is almost temperature-independent, similar to the abruptly terminated pseudogap in hole-doped superconducting cuprates[37,38]. Last but not least, the heavily S-substituted Ta$_2$Ni(Se,S)$_5$ system provides a promising platform to

study both solid state BCS-BEC crossover phenomenon[39] and critical fluctuations of low-energy phonons. Intriguingly, a potential competing order is observed in Fig. 2a for S-dopings above $p_L$, similar to the situation in iron-based high $T_c$ superconductors, where multiple orders are found competing with each other at low temperatures, including nematicity, spin-density wave, and superconductivity[40,41].

# Methods

## Sample synthesis and resistivity measurements
Single crystals of $Ta_2Ni(Se,S)_5$ were grown by the chemical vapor transfer method with iodine ($I_2$) as the transport agent. Starting materials composed of Ta powder (99.99%), Ni powder (99.99%), Se powder (99.99%), and S powder (99.99%) with a nominal molar ratio 2:1:5(1-$x$):5$x$ ($x$ is the nominal doping level of S) were fully ground and mixed together inside the glovebox. About 50 mg of iodine ($I_2$) was added to the mixture of the starting powder. The mixture was then vacuumed, back-filled with 1/3 Argon, and sealed inside a quartz tube with an inner diameter of 8 mm, an outer diameter of 12 mm, and a length of about 120 mm. The sealed quartz tube was placed horizontally inside a muffle furnace during the growth. The hot end reaction temperature was set to 950 °C, and the cold end was left in the air with the temperature stabilized at 850 °C. Long and thin single crystals were harvested by quenching the furnace in the air after one week of reaction. Excess iodine was removed from the surfaces of the crystals with ethanol. Electrical resistivity measurements were carried out on a commercial PPMS (Quantum Design) by the four-probe method with the current applied along the $a$-axis of the $Ta_2Ni(Se,S)_5$ single crystals.

## Angle-resolved photoemission spectroscopy (ARPES)
Synchrotron-based ARPES measurements were performed at the beamline BL5-2 of Stanford Synchrotron Radiation Laboratory (SSRL), SLAC, USA, and the BL 7.0.2 of the Advanced Light Source (ALS), USA. The samples were cleaved in situ and measured under the ultra-high vacuum below $3 \times 10^{-11}$ Torr. Data was collected by the DA30L analyzer. The total energy and angle resolutions were 10 meV and 0.2°, respectively. High statistics measurements were done with a signal-to-noise ratio of 100 ~ 150 at the Fermi level on the energy distribution curve taken at the Fermi momentum. This enables electronic structure restoration up to 4.5 ~ 5 $k_BT$ after Fermi function division[38,42]. Orbital content selection is based on the dipole transition matrix element effect, following the experimental geometry described in ref. 20.

## Single crystal x-ray scattering
Hard X-ray single-crystal diffraction is carried out at the energy of 44 keV at the beamline QM2 of the Cornell High Energy Synchrotron Source (CHESS). The needle-like sample is chosen with a typical lateral dimension of 100 microns, then mounted with GE Varnish on a rotating pin before being placed in the beam. A Pilatus 6M 2D area detector is used to collect the diffraction pattern with the sample rotated 365° around three different axes at a 0.1° step and 0.1s/frame data rate at each temperature. The full 3D intensity cube is stacked and indexed with the beamline software.

## First-principles DFT calculation
Ab initio calculations are performed using the Quantum ESPRESSO package[43,44]. The crystal structure relaxations are performed using the $r^2SCAN$ functional[45] with a semiempirical Grimme's DFT-D2 van der Waals correction[46]. S-doped compounds are simulated by the virtual crystal approximation (VCA), where the pseudopotentials of Se and S atoms are linearly interpolated (i.e. mixed together) according to the chemical composition. A $30 \times 30 \times 15$ k-mesh and a 100 Ry wavefunction energy cut-off were used. The electronic structure calculations are performed using the same $r^2SCAN$ functional with the relaxed structures.

## Two-band model for many-body simulations
We consider the two-band model with inter-band e-ph couplings to describe the physics in the $Ta_2Ni(Se,S)_5$ systems, whose Hamiltonian reads as[47]:

$$\mathcal{H}(E_g) = \sum_{k\sigma} \varepsilon_k^c(E_g) c_{k\sigma}^\dagger c_{k\sigma} + \sum_{k\sigma} \varepsilon_k^v(E_g) f_{k\sigma}^\dagger f_{k\sigma} + V \sum_{i,\sigma,\sigma'} \left(n_{i\sigma}^c + n_{i+1\sigma}^c\right) n_{i\sigma'}^f$$
$$+ \sum_{kq\sigma} \frac{g_q}{\sqrt{N}} \left[\left(a_q + a_{-q}^\dagger\right) c_{k+q\sigma}^\dagger f_{k\sigma} + \text{H.c.}\right] + \sum_q \omega_q a_q^\dagger a_q$$

$$(1)$$

where $c_{k\sigma}^\dagger$ ($c_{k\sigma}$) creates (annihilates) an electron at the conduction band (primarily Ta 5$d$) for momentum k and spin $\sigma$, with dispersion given by $\varepsilon_k^c$; and the $f_{k\sigma}^\dagger$ ($f_{k\sigma}$) creates (annihilates) an electron at the valence band (primarily Ni 3$d$), with dispersion given by $\varepsilon_k^v$ (Fig. 1c). The $n_{i\sigma}^c$ and $n_{i\sigma}^f$ are the density operators for the conduction and valence band, respectively. We employ the band structures fitted from the $Ta_2NiSe_5$ experiments, as reported in ref. 20, which determine the valence and conduction band dispersion for a negative band gap $E_g = -0.3$ eV. Due to the difficulty of fitting the full dispersions in the gapped $Ta_2Ni(Se,S)_5$ materials, we simplify the model using a rigid band separation controlled by $E_g$. That being said, the conduction and valence band structures for $Ta_2Ni(Se,S)_5$ read as

$$\varepsilon_k^c(E_g) = 3.25 - 1.8 \cos(k) - 0.9 \cos(2k) - 0.6 \cos(3k) + E_g/2 \quad (2)$$

$$\varepsilon_k^v(E_g) = -1.95 + 1.5 \cos(k) + 0.3 \cos(2k) + 0.1 \cos(3k) - E_g/2. \quad (3)$$

Here, $E_g = 0$ indicates the Lifshitz transition where $\varepsilon_{k=0}^c = \varepsilon_{k=0}^v$, while $E_g = 0.3$ eV reflects the situation of $Ta_2NiS_5$.

To approximate the lattice distortion in a finite-size simulation, we employ the average displacement of phonons

$$|x_{ph}| = \sqrt{\text{Tr}\left[\rho \hat{x}_{ph}^2\right]}, \quad (4)$$

where $\rho$ is the density matrix of the e-ph system and $\hat{x}_{ph} = (a_0^\dagger + a_0)/N$ is the uniform phonon displacement operator (In a finite-size system, the exact ground state does not spontaneously break symmetries, so that $\langle \hat{x}_{ph} \rangle \equiv 0$).

# Data availability
All data generated in this study have been deposited in the Figshare database with open access at the https://doi.org/10.6084/m9.figshare.24435481.

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

## Acknowledgements

The authors wish to thank A. F. Kemper, E. Demler, D. Y. Qiu, P. J. Guo, P. A. Volkov, L. I. Glazman for helpful discussions. Use of the Stanford Synchrotron Radiation Light Source, SLAC National Accelerator Laboratory, is supported by the US Department of Energy, Office of Science, Office of Basic Energy Sciences under Contract No. DE-AC02-76SF00515. M. H. and D. L. acknowledge the support of the U.S. Department of Energy, Office of Science, Office of Basic Energy Sciences, Division of Material Sciences and Engineering, under Contract No. DE-AC02-76SF00515. This research used resources of the Advanced Light Source, a US DOE Office of Science User Facility under Contract No. DE-AC02-05CH11231. Research conducted at the Center for High-Energy X-ray Science (CHEXS) is supported by the National Science Foundation (BIO, ENG and MPS Directorates) under award DMR-1829070. Work at Lawrence Berkeley National Laboratory was funded by the U.S. Department of Energy, Office of Science, Office of Basic Energy Sciences, Materials Sciences and Engineering Division under Contract No. DE-AC02-05-CH11231 within the Quantum Materials Program (KC2202) which provided the numerical simulations and within the Theory of Materials Program (KC2301) which provided the DFT calculations. The quantum many-body simulations (S.D. and Y.W.) are supported initially by U.S. Department of Energy, Office of Science, Basic Energy Sciences, under Early Career Award No. DE-SC0022874 and are completed with the support under award No. DE-SC0024524. The many-body and DFT simulations were performed on the Frontera and Stampede2 computing system, respectively, at the Texas Advanced Computing Center. The work at Yale University is partially supported by the National Science Foundation (NSF) under DMR-2239171.

## Author contributions

Y.H. and Y.W. conceived the project. C.C. carried out ARPES measurements and analysis with the assistance of Z.K., S.W., Y.H., Y.C.. Single crystals were synthesized and characterized by X.C. with guidance from R.B.. XRD measurements and analysis were performed by X.C. and Y.H.

with the help of Z.K. and J.R.. DFT calculations were performed by W.T, Z.L. with guidance from S.L.. Model calculations were performed by S.D and Y.W.. K.S. and E.N carried out preliminary STM characterization. J.R., M.H., D.L., C.J., A.B., and E.R. maintain the XRD and ARPES beamlines. C.C., W.Y, W.T. and Y.H. wrote the manuscript with the input from all authors. All authors contributed to the scientific planning and discussion.

## Competing interests

The authors declare no competing interests.
