## [Peer Review File · Nature Communications]

REVIEWER COMMENTS

Reviewer #1 (Remarks to the Author):

The authors expand on previous studies on $\text{Ta}_2\text{Ni}(\text{Se}, \text{S})_5$ by performing high-resolution ARPES and first principle calculations. Their results shed light on the electronic and structural transition evolution upon S doping in this excitonic insulator candidate family. Structural data (XRD, Raman) show a monotonic decrease of the transition temperature towards $T = 0$ with doping in this system. In contrast, resistivity data hints towards a more sudden transition at $\text{PI} \sim 70\%$ doping. Chen et al. directly resolve the electronic band structure with polarization-resolved ARPES and confirm a semimetal-to-semiconductor transition at PI as seen in resistivity. Additionally, informed by a minimum exactly solvable model, the authors argue that strong electron-phonon coupling is needed to explain the structural evolution. The authors conclude that $\text{Ta}_2\text{Ni}(\text{Se}, \text{S})_5$ substantially departs from an Excitonic insulator ground state.

The data and theoretical analysis presented in this work are of high quality. The methodology is sound. The conclusions and interpretation of the authors are well substantiated by the data and the literature. However, I cannot recommend this manuscript for publication in Nature Communication in its current form. My main concern lies in the “novelty” of these results. In my opinion, this work is currently incremental and not significant enough to the field, but I could be persuaded otherwise. In particular:

1. A discrepancy in the evolution of the electronic and lattice structure was previously reported (for example, in Ref. 27). Currently, the ARPES data is only used to confirm the temperature dependence observed in resistivity. Further discussions on the implication of the orbital evolution with doping, the gap extraction procedure, and the comparison to the DFT calculations in the main text help hone in on the importance of resolving the evolution of the electronic structure with photoemission.
2. One of the main results of this work, the fact that strong-electron phonon coupling can result in a continuous evolution of the average lattice parameters, could be emphasized more. For example, in the main text, the authors should provide limits on the Coulomb interaction for which this behavior would be observed. Including some of the data from Extended Figures 7 and 8 in Figure 4 could help.

Some other comments:

1. Can the authors include guides to the eye in Figure 2a
2. How are the error bars in Figure 3c and 3d calculated? What do they represent?

3. Following, Hattori et al. *Advanced Quantum Technologies*, 6,6 230034 (2023) can the authors comment on their sample homogeneity?

Reviewer #2 (Remarks to the Author):

Dear editor,

working through the manuscript “Anomalous excitonic phase diagram in band-gap-tuned Ta₂Ni(Se,S)₅” by C. Chen et al. I do find that the work presents one of the most insightful and clear presentations on the question of the potential excitonic insulator state in Ta₂NiSe₅ and the complicated interplay of the excitonic instability and the structural distortion.

Central methods of the manuscript ARPES and XRD to follow band structure (including orbital character) and lattice structure while tuning the bandgap of the system via S substitution of Se.

Transport measurements suggest a Lifshitz transition around 70% doping and match results from previous works. However, XRD measurements show a 2nd order transition with a very gradual change of the structural order parameter all the way to Ta₂NiS₅.

Precise ARPES measurements that allow orbital reconstruction confirm the transport measurements.

This points to a picture that goes beyond a pure bandgap driven EI and is in line with many reports that suggest strong coupling of the electrons and holes to phonons. Calculations show a gradual crossover from a dome-shaped structural order parameter behavior to a monotonic behavior when tuned from weak to strong coupling. That allows explaining already the main experimental observations.

The authors also reference to the pseudogap like phenomena reported in previous works.

As said the storyline is presented very clear and is easy to follow. Also the data is of very high quality and presented well. The only thing I miss is a bit deeper discussion of previous works that also

brought up strong phononic coupling previously; as e.g. ref [22] in the manuscript, or previous ARPES works as ref [23] describing the structural influence.

In the introduction I do miss a few early works from optics that already point to a strong phonon coupled system,

Phys. Rev. B 95, 195144 (2017) or Phys. Rev. B 98, 125113 (2018),

as well as time resolved optics that report phonon coupled order parameter dynamics:

Science Adv. 4, eaap8652 (2018) for the amplitude mode, and Science Adv. 7, eabd6147 (2021) for the phase mode.

Reviewer #3 (Remarks to the Author):

While Ta₂NiS₅, a regular semiconductor, does not undergo any phase transitions, its isostoichiometric compound, Ta₂NiSe₅, experiences an orthorhombic-to-monoclinic structural phase transition. There is an ongoing debate regarding whether this phase transition might be linked to an excitonic phase transition. In this manuscript, the authors provide a comprehensive investigation of the angle-resolved photoemission spectra of Ta₂Ni(Se,S)₅ across various levels of S-doping. Simultaneously, they examine the magnitude of lattice distortion in the monoclinic phase using x-ray diffraction. The experimental results are compared with band dispersions by density-functional theory and the phonon distortion calculated using a two-orbital effective model incorporating electron-phonon couplings and interorbital Coulombic interactions. Based on these findings, the authors conclude that the phase transition of Ta₂Ni(Se,S)₅ originates from strong electron-phonon interaction, rather than from the Coulombic interaction.

The origin of the phase transition in Ta₂NiSe₅ remains controversial. Particularly in the study of excitonic insulators, the control of gap size is crucial. Therefore, the present study that systematically investigates the band structure and lattice distortion due to S-doping is considered to be significant in this field. Additionally, the results are clearly presented in the manuscript. For these reasons, I think that this manuscript is worthy of publication in Nature Communications.

However, before endorsing it for publication, I ask the authors to address the following questions and comments:

(a)

The temperature dependence of electrical resistivity of Ta₂NiSe₅, as illustrated in Extended Data Fig. 2, shows a decrease even above the transition temperature, indicating a semiconductor-like behavior. This is consistent with the observations in Refs. [14,15]. However, when looking at the ARPES results for Ta₂NiSe₅ in Fig. 3a, it appears that in the orthorhombic phase above the transition temperature the spectra exhibit a finite value at the Fermi level, suggesting that it is metallic. What causes this inconsistency?

(b)

Related to the previous question, the ARPES results of Ta₂NiSe₅ given in Refs. [18,19] seem to present an energy gap, meaning that there are no crossing of the conduction and valence bands even at the orthorhombic phase. Could the authors explain the reason for this discrepancy?

(c)

In Fig.2a, where S-doping is large, the lattice distortion β initially increases just below the transition temperature, but decreases again as the temperature continues to decrease. Why is this the case? This peaked behavior seems to be observed with S-doping levels higher than the p_L estimated from the resistivity measurements. Is there any correlation?

(d)

What ϵ_c^k and ϵ_v^k were used in the theoretical analysis? I guess that they are the same model used in the authors' preprint [20], and the energy-level differences of valence and conduction bands were adjusted to the values derived from the DFT results provided in this manuscript. At the very least, the paper should include enough information for readers to understand the model and computational methods by solely referring to this paper.

(e)

Regarding the second line of Eq. (1), it seems that n_{σ}^f is the correct notation.

Response to Reviewers

Reviewer 1

The authors expand on previous studies on $Ta_2Ni(Se, S)_5$ by performing high-resolution ARPES and first principle calculations. Their results shed light on the electronic and structural transition evolution upon S doping in this excitonic insulator candidate family. Structural data (XRD, Raman) show a monotonic decrease of the transition temperature towards $T = 0$ with doping in this system. In contrast, resistivity data hints towards a more sudden transition at $PI \sim 70\%$ doping. Chen et al. directly resolve the electronic band structure with polarization-resolved ARPES and confirm a semimetal-to-semiconductor transition at PI as seen in resistivity. Additionally, informed by a minimum exactly solvable model, the authors argue that strong electron-phonon coupling is needed to explain the structural evolution. The authors conclude that $Ta_2Ni(Se, S)_5$ substantially departs from an Excitonic insulator ground state.

The data and theoretical analysis presented in this work are of high quality. The methodology is sound. The conclusions and interpretation of the authors are well substantiated by the data and the literature. However, I cannot recommend this manuscript for publication in Nature Communication in its current form. My main concern lies in the “novelty” of these results. In my opinion, this work is currently incremental and not significant enough to the field, but I could be persuaded otherwise. In particular:

Authors' reply:

We thank the reviewer for recognizing the high quality of our work, and for providing very constructive suggestions. We realize that our work's novelty may not be sufficiently conveyed, and would like to clarify both in the response and in our revised manuscript.

A quick summary of our work's importance and novelty is: direct determination of the temperature-bandgap phase diagram is one most straightforward ways to resolve the excitonic insulator dilemma in $Ta_2Ni(Se,S)_5$. However, the phase diagram – especially when it comes to the Lifshitz transition doping and the phase transition temperatures – is highly contested and inconsistent from previous studies, because these were inferred from measurements that are not directly sensitive to the single-particle band gap or symmetry breaking. Enabled by the necessary resolution and statistics, we use angle-resolved photoemission and XRD to directly pin down such a phase diagram. From here, we discuss the possibility of an EI ground state, how it departs from a textbook weak-coupling regime, and unexpected signatures of potential phase competitions.

The following provides more details on why our result is both novel and important to the field, which reviewers 2 and 3 also kindly agree. Reviewer 1's crucial comments do make us realize that the present manuscript need extensive revision to sharpen the novelty and message.

Solving the long-standing debate on Ta_2NiSe_5 .

The key question associated with Ta_2NiSe_5 is the long-standing debate of whether it is an excitonic insulator, or more generally, to what extent the direct electron-hole Coulomb

interaction contributes to its insulating ground state. Despite extensive research efforts over the past couple of years, an agreement is far from being reached. We use one of the most direct ways to address this issue: map out the band-gap tuned phase diagram of Ta_2NiSe_5 and examine if it is compatible with the canonical excitonic insulator prediction. As illustrated in RFig. 1, the theoretical proposal of an excitonic insulator phase - spontaneously formed excitons as a thermodynamic ground state - was predicted to peak near the Lifshitz transition between a semimetal and a semiconductor, forming a dome-like phase diagram [Phys. Rev. 158, 462 (1967), Rev. Mod. Phys. 40, 755 (1968)]. This is due to the rapidly increasing electron-hole kinetic energy (thus screening) towards the semimetal side ($E_g < 0$), and the rapidly increasing band gap that eventually negates the energy gain to form excitons on the semiconductor side ($E_g > 0$).

RFig. 1 [Fig. 1a]: Pedagogical phase diagram(s) in indirect band gap controlled excitonic insulator (EI) across the semimetal-to-semiconductor transition. The sketches on the top illustrate the evolution of the band structure as the system is tuned across the transition. Excitonic instability is supposed to be maximized around the transition point when the temperature renormalized band gap $E_g(T)=0$.

Here in Ta_2NiSe_5 , such semimetal to semiconductor Lifshitz transition can be realized via the sulfur (S) substitution over selenium (Se) atoms. Previous studies, especially resistivity [Nat. Commun. 8, 14408 (2017)] and Raman measurements [Phys. Rev. B 104, L241103 (2021)] -- while suggestive -- are not direct probes of the order parameter of the lattice and electronic symmetry breaking. This leaves room for much inconsistency in the speculated phase diagram, due to either inaccurately determined resistivity jump [Nat. Commun. 8, 14408 (2017)], or inaccurately extrapolated “fictitious” phase transition temperatures from Raman susceptibility [Phys. Rev. B 104, L241103 (2021)]. Moreover, the key to this phase diagram - the normal state band gap -- has been under debate even for the parent compound Ta_2NiSe_5 [J. Supercond. Nov. Magn. 25, 1231 (2012), Phys. Rev. B 90, 155116 (2014), Nat. Commun. 9, 4322 (2018), Phys. Rev. Res. 2, 013236 (2020), Nat. Phys. 17, 1024 (2021), Proceedings of the National Academy of Sciences 120, e2221688120 (2023)]. Thus, a bona fide phase diagram of $\text{Ta}_2\text{Ni}(\text{Se},\text{S})_5$ can only be well depicted if the two key parameters, the critical temperature (T_s) of structural phase transition and the band overlap/gap (E_g) of the electronic structure, can be precisely determined for each sample at different S-doping levels. This is why we employ both the high-resolution XRD and ARPES techniques, which are the direct probes that trace the change of lattice and electronic structures, respectively. As a result, the phase diagram of Ta_2NiSe_5 across the electronic Lifshitz transition is illustrated in RFig. 2,

which we found to be qualitatively different from the pedagogical excitonic insulator dome-like phase diagram. Therefore, our work provides **decisive evidence** that Ta_2NiSe_5 is NOT a canonical excitonic insulator system.

RFig. 2. Evolution of structural transition temperature T_s and the maximum Ta atom displacement $|X|$ as a function of S-doping in $\text{Ta}_2\text{Ni}(\text{Se}, \text{S})_5$. The gray shade illustrates the pedagogical excitonic phase diagram with only electron-hole Coulomb attraction, which peaks around the electronic Lifshitz transition (blue star).

Extending the excitonic phase diagram to the strong coupling regime

The idea of an excitonic insulator [Sov. phys. Solid state. 6, 2219 (1965), Sov. Phys. JETP 21, 790 (1965), Phys. Rev. 162, 752 (1967), Rev. Mod. Phys. 40, 755 (1968)] was proposed more than half a century ago, in dealing with a situation when the system is continuously tuned from a semimetal to a semiconductor (RFig. 3a). Besides the pedagogical dome-like excitonic insulator phase diagram (RFig. 3b), the original proposal also discussed the deviation of the excitonic phase diagram in different situations, for instance in the presence of crystal anisotropy or considerable electron-hole asymmetry (RFig. 3c-d). These phase diagrams have been used as the primary guide map in the heated search for bulk excitonic insulators.

RFig. 3 Archetypal phase diagrams of a weakly coupled system when it is tuned across semimetal to semiconductor Lifshitz phase transition. **a** The situation without the presence of any interaction. **b** The situation with finite coulomb interaction between co-existing electrons and holes, resulting in the pedagogical dome-like excitonic insulator phase diagram. **c-d** The altered EI phase diagram with crystal anisotropy or e-h asymmetry ($m_h/m_e > 100$). EI: excitonic insulator. NL: unstable lattice. WC: Wigner crystal.

However, the above discussion is only limited to the weakly coupled regime and we show that these phase diagrams can be greatly altered as the coupling strength of the system is tuned up. In the case of Ta_2NiSe_5 , combining both the first-principles and many-body calculations, we find that the ground state of the system across the Lifshitz transition is dramatically influenced by the presence of strong inter-band electron-phonon coupling (EPC), and the experimentally derived phase diagram can be well reproduced if the EPC strength in the model calculation is tuned up to the strongly coupled regime (Fig. 4b in the main text). This is in line with the independent prediction made in our previous work based on the parent compound Ta_2NiSe_5 [arXiv:2203.06817 (2022)], where the insulating states of the system are mainly attributed to the strong electron-phonon interaction. At the same time, our photoemission data also explain why the previous resistivity results (incorrectly) suggested the symmetry-breaking terminating at ρ_L , while the order parameter in fact doesn't. This revelation fundamentally dismantles one key misunderstanding that helped initiate the excitonic insulator debate in the Ta_2NiSe_5 system. Therefore, our work for the first time extends the discussion of the excitonic insulator phase diagram to the strong coupling regime (RFig. 4), providing important insight into the future study of bandgap-tuned metal-to-insulator transitions in strongly coupled quasi-1D systems.

RFig. 4 [Fig. 4c]: Sketched phase diagrams of band-gap-tuned semimetal-to-semiconductor transition in the strong (top) and weak (bottom) coupling limits. Solid lines mark the phase boundaries, above which the shades show order fluctuations [arXiv:2203.06817v2 (2022)]. The red shade on the bands illustrates the electronic states that participate in the phase transition. The horizontal dashed lines denote E_F .

In summary, our work not only clarifies the nature of the broken-symmetry state in a leading excitonic insulator system, solving the long-standing debate in the field, but also presents a generic phase diagram to help guide future investigations of band-gap-tuned excitonic systems in the strong coupling regime. We have revised our manuscript according to the reviewer's very constructive suggestions and added detailed text descriptions in the supplementary information to increase the readability of our work. As such, we sincerely hope our revised manuscript would merit publication in *Nature Communication*.

1. A discrepancy in the evolution of the electronic and lattice structure was previously reported (for example, in Ref. 27). Currently, the ARPES data is only used to confirm the temperature dependence observed in resistivity. Further discussions on the implication of the orbital evolution with doping, the gap extraction procedure, and the comparison to the DFT calculations in the main text help hone in on the importance of resolving the evolution of the electronic structure with photoemission.

Authors' reply:

We thank the reviewer for raising the important question regarding the essential role of the high statistics ARPES measurements in our work. The misunderstanding of ARPES results are mainly used to confirm the temperature dependence from resistivity may be due to our discussions of the pseudogap effect in the context of an earlier work (new Ref. 20), which is not our main result. We emphasize that this work's focus is on the S-doping axis, and the ARPES data does *not* simply confirm the resistivity measurements. In fact, the resistivity-derived phase boundary (e.g. Ref. 27) is at odds with that directly determined from XRD and ARPES. Here, ARPES results provide a way to reconcile this nontrivial discrepancy (presence of an electronic pseudogap and a Lifshitz transition), and further directly quantify the normal state band gap E_g - a key parameter defining the EI phase diagram. It should also be noted that Ref. 27 considers the Lifshitz transition to occur at the Se-parent compound, and ~70%

doping is the end-point of the broken-symmetry phase. This is substantially different from our results, and we contend that these earlier speculations about the phase diagram were incorrect.

As stated in the previous section, in order to map out the phase diagram of the $\text{Ta}_2\text{Ni}(\text{Se},\text{S})_5$ family, and to resolve the long-standing debate of whether it is an excitonic insulator, we must directly quantify two crucial parameters at different S-doping levels: the temperature of structural phase transition T_s and the size of electronic band overlap/gap E_g . The latter is, and can only be, directly achieved through high-temperature ARPES or scanning tunneling spectroscopy measurements, which are the most direct techniques for visualizing electronic structure and energy gaps. In addition, for the negative band gap regime, being able to resolve the electron momentum is also indispensable. Therefore, the ARPES result is the key to providing the horizontal axis (E_g) of the phase diagram. Moreover, the very high statistics and sufficient energy resolution of our measurement ensure reliable spectral analysis such as the Fermi Dirac division (to restore intensity above Fermi level up to $\sim 4.5 k_B T$), energy normalization (to compensate for pseudogap intensity depletion), and combination of polarizations (to ensure a comprehensive understanding of orbital contents), which are in turn crucial to ensure the accuracy of the extracted E_g .

As pointed out by the reviewer, the ARPES result also shows other important information on the S-doping dependence of Ta_2NiSe_5 . For instance, the pseudogap state, where the intensity around the Fermi level is strongly suppressed [details see [arXiv:2203.06817v2 \(2022\)](https://arxiv.org/abs/2203.06817v2)], is persistently observed on the semimetal side, indicating the presence of strong lattice fluctuation above T_s . This aspect – while not a main focus of this manuscript – is reflected in the phase diagram shown in Fig. 4c of the main text. Moreover, being able to resolve the orbital content also helps clearly identify the critical doping at which the conduction-valence band hybridization disappears, which is essential in determining the Lifshitz doping as well as the nature of the broken-symmetry state.

We agree with the reviewer that these aspects were insufficiently discussed in the original text. In order to clearly convey the importance and distinctively new information revealed by ARPES measurements, we have made extensive edits both in the main text and supplementary materials accordingly.

Changes made:

- **We now clarify that previous literature contains vast discrepancies regarding the temperature-bandgap phase diagram in the $\text{Ta}_2\text{Ni}(\text{Se},\text{S})_5$ system, and that our results not only provide a most direct, definitive answer, but can also reconcile the discrepancies among literature.**
- **We now emphasize how the orbital resolved ARPES result, as well as its comparison with DFT, help convincingly determine the critical dopings and the associated band hybridizations in the anomalous excitonic phase diagram.**

“In $\text{Ta}_2\text{Ni}(\text{Se},\text{S})_5$, this question can be addressed since the value of the direct band gap can be effectively tuned by the isovalent sulfur substitution of selenium [Nat. Commun. 8, 14408 13 (2017), Phys. Rev. B 106, 075148 (2022)]. However, the phase diagram of $\text{Ta}_2\text{Ni}(\text{Se},\text{S})_5$ -- especially where the broken symmetry phase ends and where the normal state crosses from a gapless to a gapped electronic structure with S-doping -- has seen vastly contradicting results [J. Less-common Met. 116, 51 (1986), Nat. Commun. 8, 14408 13 (2017), Phys. Rev. B 104, L241103 (2021)], mainly due to the lack of direct probes of the single-particle band gap and the order parameter of the broken symmetry. A general impact of S-substitution on the low-energy electronic structure can be revealed through the normal-state (defined here as the high-temperature symmetric structure phase) resistivity measured along the chain direction (see Fig. 1b), which slowly increases until an abrupt upturn around 70% doping.”

“We first track the evolution of the broken symmetry phase boundary in $\text{Ta}_2\text{Ni}(\text{Se},\text{S})_5$ with direct measurements of the structural order parameter. In bulk excitonic insulators, exciton condensation is always tied to a simultaneous lattice distortion [Rev. Mod. Phys. 40, 755 (1968)].”

“Previous studies have reported conflicting values of ρ_L from resistivity, Raman measurements [Nat. Commun. 8, 14408 13 (2017), Phys. Rev. B 104, L241103 (2021)]. Here, we conduct high-resolution ARPES measurement, which is a direct probe of electronic structure, to reveal the evolution of band overlap/gap of $\text{Ta}_2\text{Ni}(\text{Se},\text{S})_5$ across the semimetal-semiconductor phase transition. The high statistics of the measurements enable electronic state restoration up to $4.5k_B T$ above E_F (see Methods), and the orbital selectivity through polarized X-ray beam, allows the direct comparison with density functional theory (DFT) calculations, both of which help us precisely locate ρ_L . Figure 3 shows the photoemission spectra in both the normal and the broken symmetry state (also see Supplementary Note 1).”

2. One of the main results of this work, the fact that strong-electron phonon coupling can result in a continuous evolution of the average lattice parameters, could be emphasized more. For example, in the main text, the authors should provide limits on the Coulomb interaction for which this behavior would be observed. Including some of the data from Extended Figures 7 and 8 in Figure 4 could help.

Authors' reply:

We thank the reviewer for noticing this previously underdeveloped point. In the revised manuscript, we have followed the reviewer's suggestion and extended the Extended Fig. 7 and 8 (now labeled as Fig. S7 and S8) into a 3-page section in the SI as Supplementary Note 4. This section discusses in detail, the impact of the Coulomb interaction in both mean-field and many-body contexts and the limit of it in explaining the gap-tuned phase transition. Such

an impact is primarily reflected as the Hartree shift and direct gap (instead of the hybridization gap) opening on the Ta₂NiS₅ side. This is in sharp contrast to the impact of electron-phonon interaction, which can consistently address the dome-like excitonic instability at weak coupling and monotonic order parameter at strong coupling. While distinguishing the contributions from either interaction is difficult for a single material at a given doping, the order parameter evolution across the Lifshitz transition gives a clear assignment of their contributions.

Thanks to the reviewer's suggestion, we have added this very detailed discussion in SI, which we hope provides richer information and a more conclusive discussion. This section is referenced in the main text, where Fig.4 and the monotonic evolution of order parameters are discussed.

Changes made:

- **A very detailed discussion on the impact of the Coulomb interaction has been added to the SI, i.e SUPPLEMENTARY NOTE 4: IMPACT OF THE COULOMB INTERACTION. (The whole section is not copied here for conciseness)**
- **We have added a concise statement about the impact of Coulomb interaction in the main text, referencing SUPPLEMENTARY NOTE 4.**

“To examine the impact of strong electron-phonon coupling on the band-gap-tuned transition as a matter-of-principle question, we first set the inter-band Coulomb interaction to zero. **The role of this Coulomb interaction primarily lies in a Hartree shift between two bands and can be mapped to a correction of E_g (see detailed discussions in Supplementary Note 4).**”

Some other comments:

1. Can the authors include guides to the eye in Figure 2a

Authors' reply:

We thank the reviewer for the suggestion.

Changes made:

- **Guides to the eye have been added in the revised Fig. 2a.**

Revised **Fig. 2a**: Change of the inter- and intra-chain Ta-Ta bond angle $\beta - 90^\circ$ as a function of temperature and S-doping.

2. How are the error bars in Figure 3c and 3d calculated? What do they represent?

Authors' reply:

We thank the reviewer for the question regarding error bars, which was not reflected in the previous main text.

To estimate the band overlap/ band gap (E_g) in the high-temperature ARPES spectra, we fitted both the conduction and valence band dispersions. As illustrated in RFig. 5 (previous Extended Data Fig. 4, now Fig. S6), we first trace the dispersion of the bands by peak-fitting the energy distribution curves (EDCs) or momentum distribution curves (MDCs) of the ARPES spectra. Then, the conduction band is fitted with the parabolic curve (blue dashed line), and the valence band is fitted using 3 different band shapes: linear (dashed yellow line), hyperbola (red line), and DFT result (dashed orange line). The upper/lower bound of E_g comes from the fitting result of the linear/DFT band shape.

In the case of quantifying low-temperature band back-bending momentum $2k_F$, similarly, we trace the dispersion of the valence band by peak-fitting the EDCs and MDCs of the ARPES spectra. The $2k_F$ is taken from the distance between the valence band top and the error comes from the momentum resolution of the ARPES measurement.

The error bars previously plotted in the DFT results (Fig. 3d) come from the size of the k-grid employed in our band structure calculations. In the revised manuscript, we use a much finer k-grid to accurately calculate the back-bending momentum $2k_F$, and lower the error from the k-grid size to less than 0.002 \AA^{-1} , which is far beyond the level of accuracy that DFT can reliably ensure. Therefore, the error bar on the DFT calculation is removed in our revised manuscript.

RFig. 5 [Fig. S6]: **a** High-temperature photoemission spectra of 22% S-doped sample in both LH (linear horizontal) and LV (linear vertical) channels, highlighting conduction and valence band respectively. The spectra are normalized along the energy axis to remove the effect of the pseudogap state. The “+” marks are deduced from the peak fitting of energy distribution curves (EDCs) or momentum distribution curves (MDCs), which were then fitted by hyperbola to extract the size of the band overlap E_g . The upper (lower) bound of E_g is estimated from the fitting of the valence band with a linear (normalized DFT dispersion) band shape. **b** Low-temperature photoemission spectra of 22% S-doped sample in LV channel. Band dispersion is deduced from the peak fitting of EDCs or MDCs, from which the size of $2k_F$ is extracted.

Changes made:

- **A detailed description of the fitting method has been added to the SI.**

SUPPLEMENTARY NOTE 2: FITTING OF PHOTOEMISSION SPECTRA

“To estimate the band overlap/gap E_g in the high-temperature ARPES spectra, we fitted both the conduction and valence band dispersion. As illustrated in Fig. S6, the spectra were first normalized along the energy direction to compensate for the spectra weight depletion around the Fermi level, resulting from the pseudogap state [arXiv:2203.06817 (2022)]. Then, the dispersion of the bands was traced by peak-fitting the energy distribution curves (EDCs) or momentum distribution curves (MDCs) of the ARPES spectra. The conduction band is fitted with the parabolic curve (blue dashed line), and the valence band is fitted using 3 different band shapes: linear (dashed yellow line), hyperbola (red line), and DFT result (dashed orange line). The upper and lower bound of E_g comes from the fitting result of the linear and DFT band shapes, respectively. In the case of quantifying low-temperature band back-bending momentum k_F , similarly, we trace the dispersion of the valence band by peak-fitting the EDCs and MDCs of the ARPES spectra. The $2k_F$ value is taken from the distance between the valence band top and the error comes from the momentum resolution of the ARPES measurement.”

- **DFT calculation with finer grid is updated in the revised Fig. 3 and the error bar is removed accordingly.**

Revised **Fig. 3d** (right panel): Evolution of $2k_F$ extracted from DFT calculation and photoemission spectra as a function of S-doping.

3. Following, Hattori et al. *Advanced Quantum Technologies*, 6,6 230034 (2023) can the authors comment on their sample homogeneity?

Authors' reply:

We thank the reviewer for the question regarding sample homogeneity. In short, the samples in this study are found to be quite homogenous, which is ensured by careful and comprehensive characterizations. No evidence of side peaks or smearing is spotted in our XRD measurement, and the ARPES result shows a smooth evolution of electronic structure consistent with DFT predictions.

Specifically, to check the spatial homogeneity of S doping within each doped sample, we further performed EDX mapping over the sample surface, and RFig. 6 shows the results on the select samples with 64% and 72% S-doping levels. We find that the S doping level within each sample is homogeneous with an average variation of up to $\pm 2\%$ between regions of 10-20 μm lateral dimensions, which matches our smallest ARPES beam spot sizes.

RFig. 6. Spatial homogeneity of S doping in $\text{Ta}_2\text{Ni}(\text{Se},\text{S})_5$. **a (c)** SEM image of the surface of a sample with 64% (72%) S-doping. **b (d)** The false-color plot of the ratio between the intensity of sulfur K-alpha peak and selenium L peak corresponding to **a (c)**. The green numbers on **b (d)** indicate the ratio between the intensity of sulfur K-alpha peak and selenium L peak averaged in the quarters segmented by the green dashed lines. The variation of the actual S doping level determined from our EDX

measurements in each sample is typically less than 1%. For some samples, it can reach about 2%. The horizontal error bars reflecting the variation are typically smaller than the size of the symbols in the figures of the main text (such as Figs. 1b, 2b, 3c, and 4a).

Changes made:

- **The EDX result discussed above and RFig. 6 (revised Fig. S5) have been added to the SI, illustrating the homogeneity of the S-doped samples.**

SUPPLEMENTARY NOTE 1: COMPLETE DATA OF RESISTIVITY, XRD, ARPES, AND EDX MEASUREMENTS

“To check the spatial homogeneity of sulfur doping within each doped sample, we performed energy dispersive X-ray spectroscopy (EDX) mapping over the sample surface, and Fig. S5 shows the results on the select samples with 64% and 72% S-doping levels. We find that the S-doping level within each sample is homogeneous with an average variation of up to $\pm 2\%$ between regions of 10-20 microns lateral dimensions, which matches our smallest ARPES beam spot sizes.”

Reviewer 2

working through the manuscript “Anomalous excitonic phase diagram in band-gap-tuned Ta₂Ni(Se,S)₅” by C. Chen et al. I do find that the work presents one of the most insightful and clear presentations on the question of the potential excitonic insulator state in Ta₂NiSe₅ and the complicated interplay of the excitonic instability and the structural distortion.

Central methods of the manuscript ARPES and XRD to follow band structure (including orbital character) and lattice structure while tuning the bandgap of the system via S substitution of Se.

Transport measurements suggest a Lifshitz transition around 70% doping and match results from previous works. However, XRD measurements show a 2nd order transition with a very gradual change of the structural order parameter all the way to Ta₂NiS₅.

Precise ARPES measurements that allow orbital reconstruction confirm the transport measurements.

This points to a picture that goes beyond a pure bandgap driven EI and is in line with many reports that suggest strong coupling of the electrons and holes to phonons. Calculations show a gradual crossover from a dome-shaped structural order parameter behavior to a monotonic behavior when tuned from weak to strong coupling. That allows explaining already the main experimental observations.

The authors also reference to the pseudogap like phenomena reported in previous works.

As said the storyline is presented very clear and is easy to follow. Also the data is of very high quality and presented well. The only thing I miss is a bit deeper discussion of previous works that also brought up strong phononic coupling previously; as e.g. ref [22] in the manuscript, or previous ARPES works as ref [23] describing the structural influence.

In the introduction I do miss a few early works from optics that already point to a strong phonon coupled system, Phys. Rev. B 95, 195144 (2017) or Phys. Rev. B 98, 125113 (2018), as well as time resolved optics that report phonon coupled order parameter dynamics: Science Adv. 4, eaap8652 (2018) for the amplitude mode, and Science Adv. 7, eabd6147 (2021) for the phase mode.

Authors' reply:

We thank the reviewer for recognizing the high quality of our data as well as the key information delivered to the field.

Following the reviewer's suggestion, early optics works that suggest a strong phonon-coupled system have been referenced in the introduction. We also added an expanded discussion on these pioneering works suggesting strong electron-phonon coupling.

To increase the readability of our work, we also added more detailed text descriptions in the revised supplementary information.

Changes made:

- **A brief statement in the last paragraph about the consistency of our conclusion with Ta₂NiSe₅ experiments in and out of equilibrium.**

“Instead, a strong electron-phonon-coupling induced structural transition can explain the observed symmetry breaking. This conclusion is consistent with recent experimental evidence of the crucial role of phonons in the equilibrium and nonequilibrium properties of Ta₂NiSe₅ [Proc. Natl. Acad. Sci. U.S.A. 120, e2221688120 (2023), Phys. Rev. Research 2, 013236 (2020)].”

- **A detailed discussion in the SI (at the end of Supplementary Note 4”) about the symmetry reason behind the inefficiency of the Coulomb interaction, referring to previous studies in Ref. 21 and 23.**

“The inefficiency of the Coulomb interaction in forming excitons results from the small Fermi momentum and the opposite parity of the low-energy bands, which has been discovered by earlier experimental and theoretical studies [Phys. Rev. Lett. 124, 197601 (2020), Phys. Rev. Research 2, 013236 (2020)]. The mismatch of parity completely excludes the purely Coulomb-driven excitonic orders at $k = 0$. This explains the diminishment of order parameters for E_g , reflecting the reality close to the Ta₂NiS₅ side of the material. While such an exclusion is no longer exact for finite Fermi momenta, the small k_F in Ta₂NiSe₅ leads to limited excitonic instability, compared to the substantial Hartree shift caused by the Coulomb interaction [arXiv:2203.06817v2 (2022)]. In contrast, the intraband B_{2g} phonon connects the two bands directly and breaks the mirror symmetry. Thus, the coupling to lattice distortion exhibits more efficiency in forming excitons.”

- **SI is now enriched with detailed text descriptions. (The whole SI is not copied here for conciseness)**

Reviewer 3

While Ta₂NiS₅, a regular semiconductor, does not undergo any phase transitions, its isostoichiometric compound, Ta₂NiSe₅, experiences an orthorhombic-to-monoclinic structural phase transition. There is an ongoing debate regarding whether this phase transition might be linked to an excitonic phase transition. In this manuscript, the authors provide a comprehensive investigation of the angle-resolved photoemission spectra of Ta₂Ni(Se,S)₅ across various levels of S-doping. Simultaneously, they examine the magnitude of lattice distortion in the monoclinic phase using x-ray diffraction. The experimental results are compared with band dispersions by density-functional theory and the phonon distortion calculated using a two-orbital effective model incorporating electron-phonon couplings and interorbital Coulombic interactions. Based on these findings, the authors conclude that the phase transition of Ta₂Ni(Se,S)₅ originates from strong electron-phonon interaction, rather than from the Coulombic interaction.

The origin of the phase transition in Ta₂NiSe₅ remains controversial. Particularly in the study of excitonic insulators, the control of gap size is crucial. Therefore, the present study that systematically investigates the band structure and lattice distortion due to S-doping is considered to be significant in this field. Additionally, the results are clearly presented in the manuscript. For these reasons, I think that this manuscript is worthy of publication in Nature Communications.

Authors' reply:

We thank the reviewer for recognizing the importance of our work and the key message delivered to the field. We have revised our manuscript according to the reviewer's very constructive suggestions and added detailed text descriptions in the supplementary information to increase the readability of our work. Our responses to the reviewer's comments are detailed below:

However, before endorsing it for publication, I ask the authors to address the following questions and comments:

(a)

The temperature dependence of electrical resistivity of Ta₂NiSe₅, as illustrated in Extended Data Fig. 2, shows a decrease even above the transition temperature, indicating a semiconductor-like behavior. This is consistent with the observations in Refs. [14,15]. However, when looking at the ARPES results for Ta₂NiSe₅ in Fig. 3a, it appears that in the orthorhombic phase above the transition temperature the spectra exhibit a finite value at the Fermi level, suggesting that it is metallic. What causes this inconsistency?

(b)

Related to the previous question, the ARPES results of Ta_2NiSe_5 given in Refs. [18,19] seem to present an energy gap, meaning that there are no crossing of the conduction and valence bands even at the orthorhombic phase. Could the authors explain the reason for this discrepancy?

Authors' reply:

We thank the reviewer for pointing out the persistent insulating behavior of Ta_2NiSe_5 in the high-temperature orthorhombic phase. This is actually one of the main controversies in the early study of the Ta_2NiSe_5 system. While the DFT calculation predicts the semimetallic state in the orthorhombic phase, transport and early ARPES work show the insulating (band gap) behavior of the system. This seemingly 'inconsistent' behavior is caused by the existence of an electronic pseudogap state [arXiv:2203.06817v2 (2022), Phys. Rev. B 90, 155116 (2014), Nat. Phys. 17, 1024 (2021)], resulting from the strong lattice fluctuation in a wide temperature range above T_s .

RFig. 7: Temperature dependence of the single-particle gap in Ta_2NiSe_5 . **a** DFT calculation of the single-particle bands in the low-temperature monoclinic phase and the high-temperature orthorhombic phase. States from the Ta $d_{x^2-y^2}$ and Ni d_{xy} orbitals are emphasized, corresponding to the linear horizontal (LH, blue) and linear vertical (LV, red) channels in the ARPES experiments, respectively. **b** Photoemission spectra along the X- Γ -X direction in both photon polarization channels at select temperatures [arXiv:2203.06817v2 (2022)].

High-statistic ARPES data along the Γ -X direction of Ta_2NiSe_5 is present in RFig. 7b, illustrating the band structure evolution across the structural phase transition $T_s \sim 329K$. While the low-temperature spectra (monoclinic, $T < T_s$) show a pronounced single-particle gap, consistent with DFT calculation (RFig. 7a), the high-temperature electronic structure (orthorhombic, $T > T_s$) exhibits anomalous behavior. Although a continuous dispersion of conduction and valence bands is 'seemingly' recovered, a pronounced spectral weight depletion is observed around the Fermi level (± 100 meV). This can be better visualized in RFig. 8, where we take out the data taken at $T = 380$ K, and integrate the intensity of spectra along the momentum direction. Strong suppression of total intensity is evidenced in both LH and LV channels. A continuous dispersion, in consistency with DFT calculation (RFig. 7a), can only be reached if the spectra are normalized along the energy axis. Such a single-particle "gapped" spectrum on top of a metallic dispersion, without a global symmetry breaking of the system, is analogous to the pseudogap state found in high- T_c cuprates.

RFig. 8: **a** Photoemission spectra of LH and LV polarizations with Fermi Dirac function divided. The integrated density of states (DOS) is plotted on the side, illustrating the spectra weight depletion around Fermi energy (E_F) within both channels. **b** The same spectra normalized along energy axes (divided by DOS), to get rid of the effect of spectra weight depletion. Continuous dispersions of both conduction and valence bands are recovered with the fitting result overlaid [[arXiv:2203.06817v2 \(2022\)](https://arxiv.org/abs/2203.06817v2)].

The pseudogap state is observed in a wide temperature range above T_s , accounting for the persistent insulating behavior observed in transport measurement. We attribute the microscopic mechanism of the pseudogap state to the strong fluctuation of lattice above T_s , and a detailed description can be found in our previous work on Ta_2NiSe_5 [[arXiv:2203.06817v2 \(2022\)](https://arxiv.org/abs/2203.06817v2), currently under review].

Changes made:

- **Description of the pseudogap state and related references are added in the revised manuscript, when discussing the high-temperature ARPES spectra.**

“As the S-doping level increases, the Ta $5d_{x^2-y^2}$ orbital component (blue) gradually withdraws from the valence band top, which becomes dominated by the Ni $3d_{xy}$ orbital (red). Pseudogap states, the strong depletion of spectra intensity around Fermi level [[Phys. Rev. B 90, 155116 \(2014\)](https://doi.org/10.1103/PhysRevB.90.155116), [arXiv:2203.06817v2 \(2022\)](https://arxiv.org/abs/2203.06817v2)], are also observed, which account for the high-temperature insulating behavior in these semimetallic samples (see ref. [[arXiv:2203.06817v2 \(2022\)](https://arxiv.org/abs/2203.06817v2)] for detailed description). Beyond p_L ($\sim 70\%$ S-doping), the normal-state conduction and valence bands are completely separated in energy ...”

(c)

In Fig.2a, where S-doping is large, the lattice distortion $\beta \sim 90^\circ$ initially increases just below the transition temperature, but decreases again as the temperature continues to decrease. Why is this the case? This peaked behavior seems to be observed with S-doping levels higher than the p_{L} estimated from the resistivity measurements. Is there any correlation?

Authors' reply:

We also notice the abnormal behavior of beta angle at low temperatures in heavily S-doped systems. Frankly, we have not gathered enough information to fully understand the microscopic origin yet. As acknowledged in the last sentence of the manuscript “Last but not

least, the heavily S-substituted $\text{Ta}_2\text{Ni}(\text{Se},\text{S})_5$ system provides a promising platform to study both solid state BCS-BEC crossover phenomenon and critical fluctuations of low-energy phonons, especially in the presence of a potential competing order suggested in Fig. 2a for dopings above p_L ." We think such suppression of order parameters at low temperatures most likely suggests the existence of competing orders near the quantum phase transition approaching 100% S-substitution. This behavior is uncannily similar to the situation in the iron-based high T_c superconductors where multiple orders are found competing with each other at low temperatures, including nematicity, spin-density wave, and superconductivity [Nat. Commun. 5, 3711 (2014), Nat. Phys. 10, 97–104 (2014)].

We are actively looking into this question and further experiment is being performed. We have added the related references to the main text to encourage the community to collectively look into this intriguing phenomenon, thanks to the reviewer's comment.

Changes made:

- **References to iron-based high T_c superconductors have been added in the last sentence of the main text.**

"Last but not least, the heavily S-substituted $\text{Ta}_2\text{Ni}(\text{Se},\text{S})_5$ system provides a promising platform to study both solid state BCS-BEC crossover phenomenon [Phys. Rev. B 81, 205117 (2010)] and critical fluctuations of low-energy phonons. **Specifically, a potential competing order is observed in Fig.2a for S-dopings above p_L , similar to the situation in iron-based high T_c superconductors, where multiple orders are found competing with each other at low temperatures, including nematicity, spin-density wave, and superconductivity [Nat. Commun. 5, 3711 (2014), Nat. Phys. 10, 97–104 (2014)]."**

(d)

What ϵ^c_k and ϵ^v_k were used in the theoretical analysis? I guess that they are the same model used in the authors' preprint [20], and the energy-level differences of valence and conduction bands were adjusted to the values derived from the DFT results provided in this manuscript. At the very least, the paper should include enough information for readers to understand the model and computational methods by solely referring to this paper.

Authors' reply:

We apologize for missing the band structure parameters in the original manuscript and we thank the Reviewer for pointing out our negligence. Indeed, we are using the same two-band model from our previous work, which was primarily fitted from ARPES experiments reported in Ref. 20 (since the DFT results always exhibit a gapped electronic ground state). These band parameters are directly used in this paper to describe the two-band model of Ta_2NiSe_5 . The simulation of the Lifshitz transition with two bands separating from each other was realized by adding an additional bare-band gap E_g between the valence and conduction dispersions. Due to the normal-state gap and the access to only occupied states by ARPES, it is no longer practical to re-fit the band structure for each S-doping. This assumption is qualitatively consistent with DFT results in the orthorhombic phase, although the latter tends to underestimate the gap. More importantly, the lattice order parameters simulated using this model can well capture the experimentally observed doping evolution. The information about

the model and justification of this treatment has been added to the Method section of the revised manuscript.

Changes made:

- **Parameters of the two-band model and the related treatment to simulate the S-doping evolution have been added to the Methods.**

METHODS

Section: Two-band model for many-body simulations

“We employ the band structures fitted from the Ta₂NiSe₅ experiments, as reported in Ref.[[arXiv:2203.06817v2 \(2022\)](https://arxiv.org/abs/2203.06817v2)], which determine the valence and conduction band dispersion for a negative band gap $E_g=-0.3\text{eV}$. Due to the difficulty of fitting the full dispersions in the gapped Ta₂Ni(Se,S)₅ materials, we simplify the model using a rigid band separation controlled by E_g . That being said, the conduction and valence band structures for Ta₂Ni(Se,S)₅ read as

$$\varepsilon_k^c(E_g) = 3.25 - 1.8 \cos(k) - 0.9 \cos(2k) - 0.6 \cos(3k) + E_g/2$$

$$\varepsilon_k^v(E_g) = -1.95 + 1.5 \cos(k) + 0.3 \cos(2k) + 0.1 \cos(3k) - E_g/2.$$

Here, $E_g=0$ indicates the Lifshitz transition, where $\varepsilon_{k=0}^c = \varepsilon_{k=0}^v$. while $E_g=-0.3\text{eV}$ reflects the situation of Ta₂NiSe₅.”

(e)

Regarding the second line of Eq. (1), it seems that $\varepsilon_{k=0}^{\sigma}$ is the correct notation.

Authors' reply:

We thank the reviewer for spotting this mistake. The equation is now corrected.

REVIEWERS' COMMENTS

Reviewer #1 (Remarks to the Author):

The authors have satisfactorily answered my concerns. Now the manuscript conveys in a better way the importance and novelty of this work. I recommend this manuscript for publication in Nature Communications.

As a minor comment, there is a formatting issue with the references in the supplementary material.

Reviewer #2 (Remarks to the Author):

I do support publication of the manuscript. I think the authors have addressed all concerns raised by the reviewers. Reviewer 3 was already in favour of the manuscript as well.

I do not share the point of reviewer 1 that the results were of high quality but incremental in their impact. The question of the excitonic or phononic nature of the phase transitions is one of the central questions in the field of excitonic insulators in the

solid state and the authors have provided one of the most direct views on this question and the complicated excitonic phononic interplay. They can't finally decide the question unambiguously, but they can clarify a lot of inconsistencies that appear the present literature.

In the revised manuscript the authors have even more made clear that the manuscript contains novel and important information.

Thus I substantiate my previous conclusion and fully support publication.

Reviewer #3 (Remarks to the Author):

The authors have responded appropriately to the reviewers' comments and have made suitable revisions to the manuscript accordingly.

Therefore, I believe the manuscript is now ready for publication in Nature Communications in its current form.